# OpenReview forum: "DSP: Dynamic Sequence Parallelism for Multi-Dimensional Transformers"
_ICLR.cc/2025/Conference — Submitted to ICLR 2025_

### Official Review · Reviewer_CrH5 · 2024-11-01

**Soundness:** 3
**Presentation:** 2
**Contribution:** 3
**Rating:** 6
**Confidence:** 1

**Summary:**

The paper presents Dynamic Sequence Parallelism, a approach for scaling multi-dimensional transformers efficiently. DSP dynamically switches the parallel dimension between computation stages using a resharding strategy, which reduces communication overhead and simplifies implementation. DSP provides substantial performance improvements over existing sequence parallelism methods, such as Megatron-SP and DeepSpeed-Ulysses, achieving higher throughput (32.2% to 10x increase) and reducing communication volume by up to 75%.

**Strengths:**

++ The paper targets important problem for multi-dimensional transformers scalability.

++ The proposed approach effectively reduced communication overhead.

**Weaknesses:**

-- The overhead associated with resharding in DSP is not clear.

**Questions:**

DSP requires data resharding between stages, which means that the memory layout of data changes dynamically. This process can complicate memory management and may lead to increased memory fragmentation or inefficient memory use. How to solve this case in the proposed appraoch?

Could the authors provide more details on the computational and memory overhead associated with the resharding operation in DSP? How this overhead scales with an increasing number of GPUs or longer sequences?

The paper mentions altering sequence parallelism between computation stages, but it is unclear under what specific circumstances or computations this alteration becomes necessary. Could the authors clarify when and why it is necessary to change the sequence parallelism strategy between different stages of computation?

---

> ### Author Response · Authors · 2024-11-21
> **Response to reviewer CrH5**
>
> We sincerely thank the reviewer CrH5 for the valuable questions and comments. For the concerns and questions, here are our responses:
>
>
>
> **Q1: Could the authors provide more details on the computational and memory overhead associated with the resharding operation in DSP? How this overhead scales with an increasing number of GPUs or longer sequences?**
>
> **A1:** Indeed, DSP need to change memory layout of data dynamically.
>
> For memory, this usually will not cause much fragmentation as we show in Figure 9 (Line 447) because of the following reasons:
>
> * The sequence have already been splitted. Therefore there will be only a part of sequence on one device, reducing the overhead to layout change.
> * DSP changes actually less frequent than any other methods, lead to less overhead.
> * Duing training, the layout of a sinlge sequence does not affect overall memory cost because parameter and activation is much larger than it.
>
> To further clarify this, we make experiments about the memory overhead of our method:
>
> | GPU num | memory overhead |
> | ------- | --------------- |
> | 8       | 0.5%            |
> | 16      | 0.4%            |
>
> For speed, we conduct an experiment to evaluate the time of changing layout and communication:
>
> | GPU num | layout change time | communication time |
> | ------- | ------------------ | ------------------ |
> | 8       | 5.2%               | 94.8%              |
> | 16      | 4.1%               | 95.9%              |
>
> **Analysis:**
>
> * The layout changing time is much less than the communication time. But still takes 5.2% of total time.
>
> * As the sequence and gpu number get larger, the layout change time become less as communication takes more time.
>
> * To fully elimiate the layout change's time, we can overlap communication with layout change.
>
> **Improvement plan**: We will add these analysis about the memory and speed overhead of our method in the revision.
>
> **Conclusion**:
>
> * Memory: The layout changes have little impact on memory cost.
> * Speed: The layout changes will occupy 4.1%-5.2% of communication time, but can be avoided by overlapping. And it will reduce as sequence and gpu becomes larger.
>
> **Q2: The paper mentions altering sequence parallelism between computation stages, but it is unclear under what specific circumstances or computations this alteration becomes necessary. Could the authors clarify when and why it is necessary to change the sequence parallelism strategy between different stages of computation?**
>
> **A2:** Thanks for the comment!
>
> **In one sentence: you need to change the sequence parallelism strategy when your current sharding dimension requires interdependent calculation.**
>
> For exmaple, let's take spatial-temporal transformer as an example. Assume the tensor shape is `[s, t, d]`, where `s` is spatial, `t` is temporal, and `d` is hidden size.
>
> Now assume we shard the `s` dimension into `[s1:s2]`. We will have `[s1, t, d]` on first device, and `[s2, t, d]` on second device.
>
> * In temporal layers, there is no need to switch stage. Because we have complete `t`.
> * In the mlp of spatial blocks, because the calculation is independet of `s` or `t`, there is also no need to switch.
> * Only in the attention of spatial blocks, we need to calculate attention over all sequence of `s`, we need to switch to get the full `s` dimension.

---

### Official Review · Reviewer_D6Z1 · 2024-11-01

**Soundness:** 3
**Presentation:** 3
**Contribution:** 3
**Rating:** 6
**Confidence:** 3

**Summary:**

This paper introduces Dynamic Sequence Parallelism (DSP), a novel approach to sequence parallelism in multi-dimensional transformers. Unlike existing methods that shard along a single sequence dimension, DSP dynamically switches the parallel dimension based on the computation stage, utilizing an efficient resharding strategy that minimizes communication overhead. DSP’s adaptable and easy-to-implement design allows it to operate across various modules without specialized modifications. Experimental results demonstrate DSP's performance advantages over state-of-the-art methods, achieving throughput improvements of 32.2% to 10x and reducing communication volume by at least 50%. The paper’s contributions include a formal definition of DSP, a comprehensive analysis of its communication and memory efficiency, and experimental validations that highlight its scalability and ease of integration with existing distributed frameworks.

**Strengths:**

Dynamic Sequence Parallelism (DSP) is a novel approach that overcomes limitations of existing sequence parallelism by dynamically switching dimensions based on computation stages, enhancing efficiency in multi-dimensional transformers. The technical foundations are solid, with thorough mathematical definitions and a comprehensive communication and memory analysis that demonstrate DSP’s advantages over state-of-the-art methods. The clarity of the paper is bolstered by well-structured figures, tables, and logical organization, making complex concepts accessible. DSP’s adaptability and scalability in high-dimensional transformer applications make it a significant contribution, with broad relevance for scaling long sequences across various transformer architectures and distributed deep learning systems.

**Weaknesses:**

The paper could benefit from broader evaluation across additional transformer architectures, including single-dimensional applications, to strengthen its generalizability claims. A deeper analysis of DSP’s computational overhead, particularly with frequent resharding on large-scale setups, would clarify if its efficiency holds consistently as dimensions and devices scale. Practical limitations, such as handling global operations involving all sequence dimensions, are briefly mentioned but would benefit from expanded discussion or mitigation strategies. More detailed examples of the API and implementation details, especially for integration with popular frameworks, would improve usability for practitioners. Additionally, weak scaling experiments indicate challenges in inter-node communication as GPU counts increase; exploring optimized strategies for inter-node setups could further enhance DSP’s scalability.

**Questions:**

How does the resharding process impact performance in practice, especially as the number of GPUs or sequence dimensions increases? Quantifying the computational or memory overhead of these operations would clarify DSP’s efficiency as scales grow and reveal any potential trade-offs.

Appendix A.4 briefly introduces the API, but more detailed examples, including guidance for integrating DSP with popular frameworks like TensorFlow or JAX, would be helpful. Could you add further explanation on API usage and potential challenges in integrating DSP across different environments?

---

> ### Author Response · Authors · 2024-11-21
> **Response to reviewer D6Z1**
>
> We sincerely thank the reviewer D6Z1 for the valuable questions and comments. For the concerns and questions, here are our responses:
>
> **Q1: How does the resharding process impact performance in practice, especially as the number of GPUs or sequence dimensions increases? Quantifying the computational or memory overhead of these operations would clarify DSP’s efficiency as scales grow and reveal any potential trade-offs.**
>
> **A1:** It's a very good question. Let's analyze from memory and speed:
>
> For memory, this usually will not cause much fragmentation as we show in Figure 9 (Line 447) because of the following reasons:
>
> * The sequence have already been splitted. Therefore there will be only a part of sequence on one device, reducing the overhead to layout change.
> * DSP changes actually less frequent than any other methods, lead to less overhead.
> * Duing training, the layout of a sinlge sequence does not affect overall memory cost because parameter and activation is much larger than it.
>
> To further clarify this, we make experiments about the memory overhead of our method:
>
> | GPU num | memory overhead |
> | ------- | --------------- |
> | 8       | 0.5%            |
> | 16      | 0.4%            |
>
> For speed, we conduct an experiment to evaluate the time of changing layout and communication:
>
> | GPU num | layout change time | communication time |
> | ------- | ------------------ | ------------------ |
> | 8       | 5.2%               | 94.8%              |
> | 16      | 4.1%               | 95.9%              |
>
> **Analysis:**
>
> * The layout changing time is much less than the communication time. But still takes 5.2% of total time.
>
> * As the sequence and gpu number get larger, the layout change time become less as communication takes more time.
>
> * To fully elimiate the layout change's time, we can overlap communication with layout change.
>
> **Improvement plan**: We will add these analysis about the memory and speed overhead of our method in the revision.
>
> **Conclusion**:
>
> * Memory: The layout changes have little impact on memory cost.
> * Speed: The layout changes will occupy 4.1%-5.2% of communication time, but can be avoided by overlapping. And it will reduce as sequence and gpu becomes larger.
>
> **Q2: Appendix A.4 briefly introduces the API, but more detailed examples, including guidance for integrating DSP with popular frameworks like TensorFlow or JAX, would be helpful. Could you add further explanation on API usage and potential challenges in integrating DSP across different environments?**
>
> **A2:** Of course. Here is the explanation of the usage of our api:
>
> 1. Split data when inputs: split(x, dim=a)
> 2. Reshard data when the current dimension requires interdependent calculation (like attention): dynamic_switch(x, cur_shard=a, tgt_shard=b)
> 3. Gather data at the end of program: gather(x, dim=b)
>
> The challenges of applying this to different environments: We think it will be very easy to transfer to other environments. All you need to change is the `communication backend`.
>
> For example, in the reshard function, we implement through pytorch's `torch.distributed.all_to_all(x)`.
>
> If we want to transfer to JAX, we only need to replace this backend with JAX's `jax.lax.all_to_all` and define the mesh space.

---

### Official Review · Reviewer_EErB · 2024-11-01

**Soundness:** 1
**Presentation:** 2
**Contribution:** 1
**Rating:** 3
**Confidence:** 4

**Summary:**

This paper presents a method for parallelising the inference computation of multi-dimensional transformers, i.e., transformers that operate on multi-dimensional data such as video streams. In a Multi-GPU setup, the authors argue that it is sufficient to exploit the inter-sequence (e.g., inter-column and inter-row) parallelism available in multi-dimensional data instead of embarking on more advanced methods that can parallelize the processing of a single sequence across several GPUs  (referred to as embedded sequence parallelism in the paper).  The approach proposed by the authors requires all-to-all communication to reshuffle the data across GPUs after processing each dimension. The main argument is that because there is substantial inter-sequence parallelism in multi-dimensional data, there is no need to exploit intra-sequence parallelism, which leads to fewer communication rounds than the approaches that support intra-sequence parallelism.

**Strengths:**

The experiments demonstrate that exploiting inter-sequence parallelism instead of intra-sequence parallelism leads to significant performance advantages when parallelising multi-dimensional transformers.

**Weaknesses:**

1) The technical contribution is quite limited. The main contribution of the solution presented appears to be insertion of all-to-all communication rounds to re-arrange the data before switching to a different dimension. This mechanism is similar to 2D FFT processing, where transformations are applied row-wise and column-wise with transpositions in between.
2) The theoretical and the practical comparisons with the methods that focus on intra-sequence parallelism (Megatron-SP, RingAttention, and DeepSpeed-Ulysses) are not fair because:
- These comparison points are not designed to support multi-dimensional data. They are designed to support large single-dimensional sequences. This submission does not necessarily offer better solutions for supporting large single-dimensional sequences, which should be better clarified in the paper.
- The comparison points focus on both training and inference and involve optimisations for reducing the activation memory usage during training. This submission appears to be covering only the inference part without covering any activation memory minimisation techniques, which should be acknowledged in the paper, perhaps in a limitations or discussions section.
- RingAttention can be efficiently implemented on simple ring-like topologies without requiring all-to-all interconnects. It is not fair to assume an all-to-all interconnect and then compare the communication complexity per link (or per device) with a solution that assumes a ring topology, which uses fewer links. It would be great if the authors could clarify their assumptions about the network topology and justify their choice of comparison methods.
- The authors build on some assumptions made in DeepSpeed-Ulysses, which this reviewer believes to be problematic: there is no significant difference between the complexity of all-reduce and all-to-all operations when the underlying topology is all-to-all. Because this submission makes the same assumptions made by DeepSpeed-Ulysses, the communication complexity comparisons given in Table III of this submission are also questionable. In particular, the authors appear to be reporting the total communication volume for Megatron-SP and RingAttention while reporting per-device communication volume for DeepSpeed-Ulysses and their work (DSP).

**Questions:**

Why do the authors compare their work only with the techniques that exploit intra-sequence parallelism? Are there no other techniques for parallelising multi-dimensional transformers? Are there no other techniques for exploiting inter-sequence parallelism? A more comprehensive literature review and comparisons would be very useful. If such techniques do not exist, it would be valuable for the authors to explicitly state this and discuss why their approach is novel in this context.

---

> ### Author Response · Authors · 2024-11-21
> **Response to reviewer EErB (1/2)**
>
> We deeply appreciate Reviewer EErB's thoughtful feedback and questions. From your questions and doubts, we believe you are an expert in high performance computing and parallelism.
>
> **But there may be some misunderstandings regarding our work that we would like to clarify. We will try our best to address all of your concerns and questions in detail as follows.** Please don't hesitate to engage in further discussion if any points require additional clarification.
>
>
>
> **Q1: The technical contribution is quite limited. The main contribution of the solution presented appears to be insertion of all-to-all communication rounds to re-arrange the data before switching to a different dimension. This mechanism is similar to 2D FFT processing, where transformations are applied row-wise and column-wise with transpositions in between.**
>
> **A1:** Thanks for the comment. Indeed, our method is not entirely new. But as many other deep learning parallelism methods like pipeline parallelism, data parallelism, ring-attention, their ideas are all borrowed from tranditional parallelism methods.
>
> Although our method is not entirely new, we consider our contribution as follows:
>
> * First to identify the limitation of preivous sequence parallelism's limitation for multi-dimensional transformers.
> * Revisit the methods in tranditional parallelism, and apply to multi-dimensional transformers.
> * Address the important parallelism problem (many people still have not realized such parallelism now!) in many popular models like protein prediction (AlphaFold), Video generation (Sora).
> * Propose an high-level abstraction and user-friendly api based on such parallelism.
>
> **Improvement plan**: We will update more context of tranditional parallelism (like 2D FFT as you mentioned) in the revision.
>
>
>
> **Q2: Why do the authors compare their work only with the techniques that exploit intra-sequence parallelism? Are there no other techniques for parallelising multi-dimensional transformers? Are there no other techniques for exploiting inter-sequence parallelism? If such techniques do not exist, it would be valuable for the authors to explicitly state this and discuss why their approach is novel in this context.**
>
> **A2:** Thanks for the suggestion.
>
> **Why only intra-sequence parallelism?**
>
> For training:
>
> * GPU memory constraints (80GB even for NVIDIA H100) make handling long sequences challenging
> * While pipeline parallelism reduces parameter memory, activation memory remains high due to micro-batches
> * Intra-sequence parallelism is thus the only choice to overcome memory limitations for long sequences. There is no other parallel methods to reduce activation memory cost.
>
> For inference:
>
> * The need for speedup necessitates intra-sequence parallel approaches.
>
> **Conclusion 1: In the context of sequence parallel in deep learning, intra-sequence parallelism is the only choice. Therefore we only compare intra-sequence parallelism.**
>
> **Why is our method novel?**
>
> Our work is novel in this domain because:
>
> - Existing sequence parallelism methods were designed for LLMs.
> - As multi-dimensional transformers grow, direct application of LLM parallelism techniques leads to inefficiencies.
> - Although not entirely novel, we are the first to realize such parallelism, and introduce an elegant, high-level abstraction for parallelizing multi-dimensional transformers.
>
> **Conclusion 2: For this emerging area of multi-dimensional models, we are the first method to propose such an efficient parallelism.**
>
> **Improvemen plan**: Thanks for you suggestion. In the next few days, we will update more context of our method in the revision.

---

> > ### Author Response · Authors · 2024-11-21
> > **Response to reviewer EErB (2/2)**
> >
> > **Q3: The theoretical and the practical comparisons with the methods that focus on intra-sequence parallelism (Megatron-SP, RingAttention, and DeepSpeed-Ulysses) are not fair because:**
> >
> > **Q3-1: These comparison points are not designed to support multi-dimensional data. They are designed to support large single-dimensional sequences. This submission does not necessarily offer better solutions for supporting large single-dimensional sequences, which should be better clarified in the paper.**
> >
> > **A3-1:** Indeed, we cannot be applied to support large single-dimensional sequences. And we have clarified this in the limitation section (Line 478). For your convinence, we show as follows:
> >
> > `One limitation of this work is that DSP is specifically designed for multi-dimensional transformers and may not adapt well to single-dimensional ones like language models.`
> >
> > **Q3-2: The comparison points focus on both training and inference and involve optimisations for reducing the activation memory usage during training. This submission appears to be covering only the inference part without covering any activation memory minimisation techniques, which should be acknowledged in the paper, perhaps in a limitations or discussions section.**
> >
> > **A3-2:** There are two factors that affect the activation memory: communication pattern and communication time.
> >
> > **Communication pattern**: We have talked about the memory cost in section 4.2 (Line 341).
> >
> > For your convinence, we provide the summary here:
> >
> > * Megatron-SP use all-gather to collect sequences, but will lead to more memory cost because it need to store the whole sequence.
> > * Ring-Attention need to use partition and overlap to reduce communication cost, lead to more memory cost.
> > * All-to-all will not increase much attention becuase it keeps sequences splitted and don't have complex operations
> >
> > **Communication times:** It's very easy to understand that the more communciation you make, the more memory cost you will have, that's the reason why DSP is better than DS-Ulysses as shown in Figure 9 (Line 447).
> >
> > **Improvement plan**: We will update the factor of communication times into the memory analysis section in the revision.
> >
> > **Q3-3: RingAttention can be efficiently implemented on simple ring-like topologies without requiring all-to-all interconnects. It is not fair to assume an all-to-all interconnect and then compare the communication complexity per link (or per device) with a solution that assumes a ring topology, which uses fewer links. It would be great if the authors could clarify their assumptions about the network topology and justify their choice of comparison methods.**
> >
> > **A3-3**: Thanks for this valuable suggestion. Indeed, we only consider NVIDIA GPUs in this work, which is normally with an all-to-all interconnect. To make this work more general, we need to consider more kind of interconnect!
> >
> > **Improvement plan**: We will add this to the experiment setting and limitation section in the revision.
> >
> > **Q3-4:** The authors build on some assumptions made in DeepSpeed-Ulysses, which this reviewer believes to be problematic: there is no significant difference between the complexity of all-reduce and all-to-all operations when the underlying topology is all-to-all. Because this submission makes the same assumptions made by DeepSpeed-Ulysses, the communication complexity comparisons given in Table III of this submission are also questionable. In particular, the authors appear to be reporting the total communication volume for Megatron-SP and RingAttention while reporting per-device communication volume for DeepSpeed-Ulysses and their work (DSP).
> >
> > **A3-4:** Thanks for the comment. Actually, we calculate per-device communication for all methods. We have demonstrated the calculation in appendix A.2 (Line 715).
> >
> > For you convenience, we will summarize the calculation as follows:
> >
> > * Megatron-LM
> >   * In attention: all gather + reduce scatter
> >   * In mlp: all gather + reduce scatter
> >   * in one layer: 2 x all gather + 2 x reduce scatter
> >   * Total in one spatial layer and one temporal layer: 4 x all gather + 4 x reduce scatter = 4M + 4M = 8M
> > * Ring
> >   * only in temporal attention: ring for key and value = 2M
> > * Ulysses
> >   * only in temporal attention: 4 x all to all = 4M/N (as all to all only send and receive partial sequence per device)
> > * DSP (ours)
> >   * only in temporal attention: 2 x all to all = 2M/N
> >
> > If you still have any concerns, feel free to discuss with us!

---

> ### Comment · Reviewer_EErB · 2024-11-24
> **Addressing the confusion and the fairness of comparisons**
>
> Thank you for the detailed explanations.
>
> The confusion stems from the existence of two related references: Megatron-LM (Shoeybi et al., 2019) and Megatron-SP (Korthikanti et al., 2022). The authors refer to Megatron-SP in Table 3 and Megatron-LM in Figures 5, 6, 7, 8, and 10. This reviewer believes that the authors are referring to Megatron-SP in all these cases, but the difference between the two is significant.
>
> While the Megatron-LM paper (Shoeybi et al., 2019) focuses on tensor and data parallelism, the Megatron-SP paper (Korthikanti et al., 2022) introduces intrasequence parallelism. The fact that this submission does not support intrasequence parallelism cannot be understated. The intersequence parallelism contributed in this submission is essentially a form of data parallelism.
>
> While the tensor-level parallelism used by Megatron-LM (Shoeybi et al., 2019) requires two all-reduce operations per forward/backward pass, Megatron-SP (Korthikanti et al., 2022) uses more expensive all-gather and reduce-scatter operations to support intrasequence parallelism. On the other hand, to support inter sequence parallelism, this submission uses all-to-all operations that are similar to all-reduce operations in terms of their communication complexity.
>
> The fairness of comparing the methods proposed in this submission with methods that support intrasequence parallelism (Megatron-SP and DS-Ulysses) remains questionable. This concern applies to both the theoretical and the experimental comparisons.
>
> For example, the authors refer to the “Accumulated Sequence Length” in Figure 5, which is the product of the temporal sequence length, which ranges between 128 and 1024, and the spatial sequence length, which is set to 4096. Such sequence lengths could be too small for Megatron-SP and DS-Ulysses to be effective, and thus lead to unfavourable results for these two methods.
>
> The reviewer suggests the following to produce more fair comparisons:
> 1) Increase the spatial sequence length from 4096 up to 1M and repeat the experiments.
> 2) Include comparisons with Megatron-LM (Shoeybi et al., 2019) in addition to Megatron-SP (Korthikanti et al., 2022) in Table 3 and in Figures 5, 6, 7, 8, and 10.
>
> References:
> - Mohammad Shoeybi, Mostofa Patwary, Raul Puri, Patrick LeGresley, Jared Casper, and Bryan Catanzaro. Megatron-LM: Training multi-billion parameter language models using model parallelism. ArXiv, abs/1909.08053, 2019.
> - Vijay Anand Korthikanti, Jared Casper, Sangkug Lym, Lawrence C. McAfee, Michael Andersch, Mohammad Shoeybi, and Bryan Catanzaro. Reducing activation recomputation in large transformer models. ArXiv, abs/2205.05198, 2022.

---

> > ### Comment · Reviewer_EErB · 2024-11-24
> > **One last question**
> >
> > Could the authors justify the use of multidimensional transformers instead of using spatial transformers alone by providing accuracy comparisons between the two?

---

> ### Author Response · Authors · 2024-11-25
> **Response to reviewer EErB (3)**
>
> **Q1: The confusion and the fairness of comparisons.**
>
> **A1:** Thanks for your valuable advice. We do the following improvements to produce more fair comparisons:
>
> **(1)** We update all confusion terms in our work about Megatron-LM and Megatron-SP, where we actually means Megatron-SP.
>
> And we will further check all details in the paper to avoid confusion.
>
>
>
> **(2)** Increase the spatial sequence length from 4096 up to 1M and repeat the experiments.
>
> We think your main concerns are:
>
> 1. Will different tensor shape affect communication speed?
> 2. How efficent are different communciation?
>
> To answer the question, we make the following experiments on 8 NVIDIA H100 GPUs with fully-connected NVLink. All communication operaters are from pytorch nccl backend.
>
> Note that `/8` means the sequence is sharded on 8 GPUs for all-to-all (DSP, Ulysses) and all-gather + reduce-scatter (Megatron-SP). But it can not be sharded for all-reduce (Megatron-LM).
>
> | tensor shape        | communication               | latency (ms) |
> | ------------------- | --------------------------- | ---------- |
> | [256/8=32, 1024, 1152] | all-to-all                  | 312        |
> | [256, 1024, 1152]   | all-reduce                  | 456        |
> | [256/8=32, 1024, 1152] | all-gather + reduce-scatter | 1401       |
> | [8/8=1, 32768, 1152]  | all-to-all                  | 311        |
> | [8, 32768, 1152]    | all-reduce                  | 456        |
> | [8/8=1, 32768, 1152]  | all-gather + reduce-scatter | 1404       |
>
> And in total:
>
> | method      | communication               | communication times per layer (two 1d blocks) | total speed (ms) |
> | ----------- | --------------------------- | -------------------------------- | ---------------- |
> | DSP         | all-to-all                  | 2                                | 614              |
> | DS-Ulysses  | all-to-all                  | 4                               | 1228              |
> | Megatron-LM | all-reduce                  | 4                                | 1824             |
> | Megatron-SP | all-gather + reduce-scatter | 4                                | 5616             |
>
> Analysis:
>
> 1. Different tensor shape will not affect communication speed.
>
>    But why long sequence can reduce communication time **ratio**?
>
>    Because the computation time of attention will be extremely long, making communication time less important (<3%).
>
> 2. all-to-all is the most efficient due to
>
>    1. Less input sequence size (only $\frac{1}{world\\_size}$ of allreduce) due to the advantages of this kind of parallelism.
>    2. Less communication volume (but the bandwidth is not as good as all-reduce's ring operation).
>    3. Less communication times.
>
> **(3)** Include comparisons with Megatron-LM (Shoeybi et al., 2019) in addition to Megatron-SP (Korthikanti et al., 2022) in Table 3 and in Figures 5, 6, 7, 8, and 10.
>
> We conduct all experiments on Megaton-LM. And all results are updated the in pdf revision. For your convenience, here is the summary:
>
> 1. Due to better communication efficiency, Megatron-LM can outperform Megatron-SP in scaling experiments, but still worse than DSP and DS-Ulysses.
> 2. But due to larger memory cost, Megatron-LM may also be worse than Megatron-SP due to larger sequence parallel size in end-to-end experiments.
>
> **Improvement plan: All modifications have been updated in the latest pdf revision.**

---

> > ### Comment · Reviewer_EErB · 2024-11-25
> >
> > Many thanks for the clarifications.
> >
> > The complexity results reported in Table 3 for Megatron-LM does not appear to be correct: where 8M is coming from is not clear to this reviewer. By revisiting the Figure 4 of Shoeybi et al., the authors could see that only two all-reduce operations are performed per layer during a forward pass (same as the backward pass). So, the complexity per transformer layer is 2M. And the communication complexity per transformer layer and per device is 2M/N (N is the number of devices). The authors should include Megatron-LM in their Figure 10 to make this clear and emphasise that they are reporting the communication volume per device.
> >
> > Otherwise, I had asked the authors to increase the size of one of the sequences as much as possible because I believed that this is the scenario (large sequences) Megatron-SP and DS-Ulysses are optimised for and, in such a scenario, these two solutions could significantly outperform the solution proposed by the authors. Could the authors comment on that? Why is it not possible to go beyond a spatial sequence length of 32768 in the experiments?

---

> ### Author Response · Authors · 2024-11-25
> **Response to reviewer EErB (4)**
>
> **Q2: Could the authors justify the use of multidimensional transformers instead of using spatial transformers alone by providing accuracy comparisons between the two?**
>
> **A2**: Thanks for the question.
>
> Normally for multi-dimensional data, if we do not use multi-dimensional transformers, we will reshape the tensor to a 1d sequence and use single-diemnsion transformer (we cannot use only spatial transformer because the there must be some interactions between different dimenisons e.g., spatial and temporal).
>
> So we will further explain why people choose multi-dimensional transformers over single-dimensional transformer for mult-dimensional data. Due to limit of discussion period, we cannot run full training experiments to evaluate accuracy. So we provide some evidence mainly from recent literature.
>
> For your convenience, we summarize the key point as follows:
>
> 1. In axiel attention (used by AlphaFold, one of the earlist multi-dimensional attention): `The experiments demonstrate that RCCA(its method) captures full-image contextual information in less computation cost and less memory cost.`
>
>    paper link: https://arxiv.org/pdf/1811.11721 (Page 12, the first paragraph below Table 9)
>
> 2. In Latte (video generation model), they compare the performance of multi-dimensional and single-dimensional transformers, and they find the multi-dimensional transformer has the best performance with less computational costs.
>
>    paper link: https://arxiv.org/pdf/2401.03048v1 (Page 4 Figure 2, and Page 10 Figure 6-d)
>
> 3. For OpenSora (multi-dimensional) and OpenSoraPlan (single-dimensional), they have simialr datasets and training strategies, only differ in model architectures.
>
>    The results is OpenSora (multi-dimensional)  has better video quality (1.76% higher score) and better speed (1.73x faster) than OpenSoraPlan (single-dimensional), according to the vbench leaderboard (https://huggingface.co/spaces/Vchitect/VBench_Leaderboard).
>
>    Note its still a little unfair because some minor differences, but can be an example.
>
> **Conclusion**: Multi-dimensional transformers have **better quality and efficiency** than single-dimensional transformers for **multi-dimensional data**.

---

> ### Author Response · Authors · 2024-11-25
> **Response to reviewer EErB (5)**
>
> Thanks for the questions!
>
> **Q1: incorrect communication volume**
>
> **A1:** In this paper, we refer one layer as two 1d blocks: `In a 2D-Transformer, there is one transformer block for each sequence
> dimension per layer, resulting in two transformer blocks per layer.` (Line 309)
>
> So the communication volume for Megatron-LM and Megatron-SP should be communication per block x 2 = 8M
>
> And the total communication times in last reply for DSP and DS-Ulysses is actually wrong (fixed now), so sorry for our mistake!
>
> Both DS-Ulysses and DSP need to communicate in only one block with in a layer (two 1d blocks) (please refer to Fig10 for details). And each time, DSP requires 2 all-to-all and DS-Ulysses requires 4 all-to-all. So their communciation volume per-layer (two 1D blocks) are 2M/N and 4M/N, which correspond to the Table 3.
>
> Therefore, the communication volume in Table 3 is correct, but exists some confusions needed to be improved. Thanks for pointing out the problem!
>
> **Improvement plan**:
> * We acknowledge the term `per layer` is a bit confused, we have made it clearer for **everywhere this term occurs** in the pdf revision. (Line 326, Line 310 and Line 723).
> * Clarify the communication is per device on Line 316, 326 and 724.
> * Update Megatron-LM workflow in Figure 10.
>
> **Q2: increase the size of one of the sequences as much as possible**
>
> **A2**: Sure, here are results of 8M length sequence. Due to memory limit, we use float16 for all tensors.
>
> | tensor shape        | communication               | speed (ms) |
> | ------------------- | --------------------------- | ---------- |
> | [8192/8=1024, 1024, 1152] | all-to-all                  | 1452        |
> | [8192, 1024, 1152]   | all-reduce                  | 2103        |
> | [8192/8=1024, 1024, 1152] | all-gather + reduce-scatter | 12977       |
> | [8/8=1, 1048576, 1152]  | all-to-all                  | 1453        |
> | [8, 1048576, 1152]    | all-reduce                  | 2103        |
> | [8/8=1, 1048576, 1152]  | all-gather + reduce-scatter | 12977       |
>
> And in total:
>
> | method      | communication               | communication times per layer (two 1D blocks) | total speed (ms) |
> | ----------- | --------------------------- | -------------------------------- | ---------------- |
> | DSP         | all-to-all                  | 2                                | 2914              |
> | DS-Ulysses  | all-to-all                  | 4                                | 5828              |
> | Megatron-LM | all-reduce                  | 4                                | 8412             |
> | Megatron-SP | all-gather + reduce-scatter | 4                                | 51908             |
>
> Analysis:
> * Megatron-LM, Megatron-SP and DS-Ulysses actually use same strategy for extreme long sequences according to both their official implementation and paper.
> * Our strategy is still the best.
>
> **We are really thankful for your questions and suggestions, which help us address confusion, improve clarity, and also fix some problems. If you still have any concern, please feel free to ask!**

---

> > ### Comment · Reviewer_EErB · 2024-11-26
> >
> > Dear authors,
> >
> > I had asked you to report the end-to-end performance when using larger sequences. As far as I understand, what you are reporting above is only the communication latency. Please note that the complexity of attention mechanisms increases quadratically with the sequence (context) length and that will dominate the execution time when the sequences are large enough. Megatron-SP and DS-Ulysses are designed to alleviate this problem by exploiting intra-sequence parallelism. However, such effects will not be visible unless you perform end to end measurements on larger sequences (contexts). Because you have not performed end-to-end measurements on large sequences, it is difficult to conclude that your method is the best for multidimensional data.
> >
> > Thank you for updating Figure 10. However, it is not difficult to see that Megatron-LM is performing fewer collective-communication rounds than Megatron-SP just by looking at your figure. So, I am afraid that your evaluation of the communication complexity of Megatron-LM in Table 3 is still incorrect.
> >
> > Furthermore, I assume you are evaluating the inference performance in all your experiments. However, you are stating that only for Figure 7 and Figure 8. You should make it clear from the beginning that you are evaluating only the inference performance.

---

> ### Author Response · Authors · 2024-11-26
> **Response to reviewer EErB (6)**
>
> Thanks for the question!
>
> **Q1: Conduct end-to-end performance for long sequences.**
>
> **A1:** The reason why we only show you the communication performance is because:
> * **The compuation remains the same regardless of different intra-sequence parallelism.**
> * According to our experiments, the computation of 1M sequence will take >98% time. In that case, there is no meaning to compare parallelism.
>
> Assume the multi-dimensional sequence shape is `[sequence1, sequence2, head_num, head_dim]`, and we will do attention on `sequence1`. What Megatron and DS-Ulysses do is to seperate the `head_num` dim of the sequence. And DSP will seperate the `sequence2` dim.
>
> But neither algorithm changes the internal logic of attention because both `head_num` and `sequence2` is independent of attention computation. The difference of each sequence parallel is only seperating different dimension, leading to different communication cost, but the computation per-device **remains the same all time**.
>
> Therefore, I think only the communication differences of these parallelism matter because **the computation is excatly the same**.
>
> **Q2: incorrect communication volume.**
>
> **A2:** We list the calculation the communication volume as follows. We're 100% sure its correct and is also corresponding to Table 3 and Figure 10. Please point out if there is any problem.
>
> * Megatron-LM
>   * In attention: allreduce
>   * In mlp: allreduce
>   * in one block: 2 x all reduce
>   * Total in one spatial block and one temporal block: 4 x allreduce = 4 x 2M = 8M
> * Megatron-SP
>   * In attention: all gather + reduce scatter
>   * In mlp: all gather + reduce scatter
>   * in one block: 2 x all gather + 2 x reduce scatter
>   * Total in one spatial block and one temporal block: 4 x all gather + 4 x reduce scatter = 4M + 4M = 8M
> * Ring
>   * only in temporal attention: ring for key and value = 2M
> * Ulysses
>   * only in temporal attention: 4 x all to all = 4M/N (as all to all only send and receive partial sequence per device)
> * DSP (ours)
>   * only in temporal attention: 2 x all to all = 2M/N
>
> **Q3**: Are all experiments for inference?
>
> **A3**: We clearly label all inference experiments with `inference` (only Figure 8). All other experiments (all except Figure 8) are for training. We will further clarifiy this.

---

> > ### Author Response · Authors · 2024-11-26
> > **Response to reviewer EErB (7)**
> >
> > As you asked, we have conducted the experiment on long sequences of shape `[8, 1048576, 1152]` in fp16.
> >
> > | method      | communication time (s) | computation time (s) |
> > | ----------- | ---------------------- | -------------------- |
> > | DSP         | 2.9                    | 362.1                |
> > | DS-Ulysses  | 5.8                    | 362.2                |
> > | Megatron-LM | 8.4                    | 365.6                |
> > | Megatron-SP | 51.9                   | 365.6                |

---

> ### Comment · Reviewer_EErB · 2024-11-28
> **Incorrect communication volume**
>
> Dear authors,
>
> There are multiple errors in the communication volume you are reporting in Table 3 for Megatron-LM. These errors have not been fixed despite all my requests.
> - Your Figure 10 correctly shows that the spatial block uses only two all-reduce operations (one for attention and one for mlp). The temporal block also uses two all-reduce operations (one for attention and one for mlp). So, "only four" all-reduce operations in total for Megatron-LM. In Figure 10, one can also see that Megatron-SP performs eight collective communication operations.
> - Just like the all-to-all operations, all-reduce operations can be parallelised, which the authors appear to be unaware of. Then, the communication complexity per device should be divided by N. This is what I have been saying since my first comment.
> - The overall communication complexity for the Megatron-LM would then be 4M/N and not 8M.
>
> Furthermore, as I suspected, the results you posted in your responses show that the DSP approach does not meaningfully outperform the remaining methods when the spatial sequence length is high (e.g., in the order of 1M). So, the DSP approach leads to significant performance improvements only in certain regimes (e.g., when the spatial sequence length is constrained). To prove the value of your approach, you would ideally also perform accuracy analyses under these different settings and show that shorter spatial sequences would also lead to a higher accuracy, etc.
>
> In summary, the reviewer believes that the theoretical analysis is still incorrect and the experimental evaluation is still incomplete.

---

> ### Author Response · Authors · 2024-11-29
> **Response to reviewer EErB (8)**
>
> Thanks again for the some many responses from reviewer EErB. Thank you for spending so much time on the discussion! We think there are still questions that remain uncertain for you like communication volume. We try to explain as detailed as we can in this reply.
>
> **Q1: Incorrect communciation volume**
>
> **A1:** The communication volume for allreduce is 2M. Because allreduce in NCCL is achieved by one all-gather and one reduce-scatter, and do some extra communication optimization. And I think there is not doubt that all-gather and reduce-scatter's communication volume is M.
>
> The reason why all-to-all only has M/N communication is because:
>
> * **The sequence to commute is naturally smaller** as its **parallel pattern in transformer model determines (VERY IMPORTANT!!!! ITS THE KEY OF ALL-TO-ALL!)**.
>
> * Its input tensor shape is naturally shard `[1024, 1024, 1152]` while all-reduce must commute the whole sequence `[8192, 1024, 1152]` due to the unique communication pattern in transformer.
>
> * **And all communication is implemented with overlapping in NCCL, so we do not count overlap for any communication.**
>
>
> | tensor shape        | communication               | speed (ms) |
> | ------------------- | --------------------------- | ---------- |
> | [8192/8=1024, 1024, 1152] | all-to-all                  | 1452        |
> | [8192, 1024, 1152]   | all-reduce                  | 2103        |
> | [8192/8=1024, 1024, 1152] | all-gather + reduce-scatter | 12977       |
>
> So the final communication is:
> | communication type        | communication  volume |
> | ------------------- | --------------------------- |
> | all-reduce        | 2M |
> | all-gather        | M |
> | reduce-scatter       | M |
> | all-to-all       | M/N |
>
> The above communication volume is supported by a lot of papers including https://arxiv.org/pdf/1910.02054 (page 14), https://arxiv.org/pdf/2305.13525 (page 16) and https://arxiv.org/pdf/2309.14509 (page 4). We believe its already well proofed.
>
> Therefore, the communication volume for each parallel method is
>
> | communication type        | communication  volume |
> | ------------------- | --------------------------- |
> | Megatron-LM        | 4 x all-reduce = 4 x (2M) = 8M |
> | Megatron-SP       | 4 x (all-gather + reduce-scatter)= 4 x ( M + M ) = 8M |
> | DS-Ulysses       | 4 x all-to-all = 4 x (M/N) |
> | DSP      | x x all-to-all = 2 x (M/N) |
>
> **Q2: DSP approach does not meaningfully outperform the remaining methods when the spatial sequence length is high.**
>
> **A2**: As we list above, DSP outperforms all existing methods by at  least 50\% less communication cost. But the communication cost doesn't dominate in such case.
>
> We believe this performance can solve your concern about `Megatron-SP and DS-Ulysses are designed to alleviate this problem by exploiting intra-sequence parallelism. However, such effects will not be visible unless you perform end to end measurements on larger sequences (contexts).`
>
> We believe it's not improper to deny the advantage of our method due to this extreme case where all parallel methods cannot contribute much. And such sequence length rarely appears in multi-dimensional data.

---

### Official Review · Reviewer_b8Jp · 2024-11-03

**Soundness:** 3
**Presentation:** 3
**Contribution:** 2
**Rating:** 6
**Confidence:** 3

**Summary:**

The paper presents Dynamic Sequence Parallelism (DSP), a novel method to scale multi dimensional transformers with long sequence length efficiently. The proposed solution dynamically switches the parallel computation dimension and minimizes communication costs and enhances adaptability. The experimental results show that communication reduction, and throughput improvement compared to state-of-the-art embedded sequence parallelism methods.

**Strengths:**

1. The paper is well organized and easy to follow.
2. The illustration with the tensor shape helps the understanding of the proposed solution.
3. Applicability to the emerging ND transformer.
4. Easy to use API design.

**Weaknesses:**

1. Can you please show the breakdown for communication overhead/actual computation etc?
2. Can you please show the performance with a larger parameter/multi axis model?

**Questions:**

Please see the weakness section.

---

> ### Author Response · Authors · 2024-11-21
> **Response to reviewer b8Jp**
>
> We sincerely thank the reviewer b8Jp for the valuable questions and comments. For the concerns and questions, here are our responses:
>
>
> **Q1: Can you please show the breakdown for communication overhead/actual computation etc?**
>
> **A1:** Of course! We will breakdown for communication overhead/actual computation for DSP under weak scaling condition using Transformer-2D 3B:
>
> | GPU num | computation time | communication time |
> | ------- | ---------------- | ------------------ |
> | 8       | 90.1%            | 9.9%               |
> | 16      | 75.2%            | 24.8%              |
>
> **Analysis**: The communication costs grows significantly for inter-node communication even for our method. Highlight the necessity of efficient parallelism.
>
> **Improvement plan**: It's a good perspective to analyze communication cost. We will add this to the revision of our work!
>
> If you need more experiments for each parallel or model, please tell us!
>
> **Q2: Can you please show the performance with a larger parameter/multi axis model?**
>
> **A2**: We conduct an end-to-end analysis for DSP with a larger parameter model with 3B, 7B and 13B using 1M tokens.
>
> | GPU num | model size | TFLOPs per GPU |
> | ------- | ---------- | -------------- |
> | 16      | 3B         | 242.20         |
> | 16      | 7B         | 262.29         |
> | 16      | 13B        | 259.77         |
>
> Analysis:
>
> * When scale from 3B to 7B, although the communication cost increase, the performance grows because the compuation density increases more.
> * When scale from 7B to 13B, the benefits of denser computation is marginal. The total throughput decrease because of more communication cost.
>
> **Improvement plan**: We will add this experiment in the appendix in the revision of our work.

---

> > ### Comment · Reviewer_b8Jp · 2024-11-24
> >
> > Thanks for the explanation and clarification! That resolves most of my concerns.

---

### Official Review · Reviewer_k5oB · 2024-11-04

**Soundness:** 3
**Presentation:** 3
**Contribution:** 3
**Rating:** 6
**Confidence:** 3

**Summary:**

This paper proposes to improve the efficiency of multi-dimensional transformers for long sequences via dynamic sequence parallelism (DSP). Unlike the conventional data parallelism, model parallelism, pipeline parallelism, etc. which are designed only within a single dimension and have limited flexibility, DSP can adaptively switch between dimensions and therefore, can minimize the communication overhead. The evaluation results show that DSP can improve the E2E throughput by 10x and at least 50\% communication. The authors also make DSP a user-friendly API that can be easily integrated into the existing transformer training/inference frameworks.

**Strengths:**

+ This paper targets an important topic -- the low efficiency of transformers on long sequences, and proposes effective solution that yields significant end-to-end throughput improvement;
+ Well written paper with clear logic, the illustration and presentation are helpful for understanding the paper's idea.

**Weaknesses:**

- The evaluation section can be further elaborated. I assume that authors are evaluating the inference tasks only. If yes, this should be explicitly pointed out. Also, the discussions for DSP on training phase are missing. For example, will DSP affect accuracy or the training convergence cycles?
- Extending the experiments on more configurations could be beneficial. See questions for details.

**Questions:**

- How does DSP impact training phase? Will it take longer to converge with DSP?
- How does DSP perform on the larger models with more parameters?
- If the experimental environment changes, for example, with less or more GPUs, how will DSP perform over the other works?
- Will DSP perform better or worse for relatively shorter sequence length?

---

> ### Author Response · Authors · 2024-11-21
> **Response to reviewer k5oB**
>
> We sincerely thank the reviewer k5oB for the valuable questions and comments. For the concerns and questions, here are our responses:
>
>
> **Q1: How does DSP impact training phase? Will it take longer to converge with DSP?**
>
> **A1**: Thanks for the comment. The answer is no, DSP will not change the results of model.
>
> **Q2: How does DSP perform on the larger models with more parameters?**
>
> **A2:** We conduct an end-to-end analysis for DSP with a larger parameter model with 7B and 13B using 1M tokens.
>
> | GPU num | model size | TFLOPs per GPU |
> | ------- | ---------- | -------------- |
> | 16      | 3B         | 242.20         |
> | 16      | 7B         | 262.29         |
> | 16      | 13B        | 259.77         |
>
> Analysis:
>
> * When scale from 3B to 7B, although the communication cost increase, the performance grows because the compuation density increases more.
> * When scale from 7B to 13B, the benefits of denser computation is marginal. The total throughput decrease because of more communication cost.
>
> **Improvement plan**: We will add this experiment in the appendix in the revision of our work.
>
> **Q3: If the experimental environment changes, for example, with less or more GPUs, how will DSP perform over the other works?**
>
> **A3**: Thanks for the comment. In the strong scaling and weak scaling experiments (Figure 6 and 7). We show how DSP performs compared with baseline when using different GPUs.
>
> For your convience, we summarize the results as follows:
>
> * The more GPUs there are, the better DSP will be compared with baseline.
> * The more challenging the environments are, DSP will be better.
>
> **Q4: Will DSP perform better or worse for relatively shorter sequence length?**
>
> **A4**: Thanks for the comment. In the end-to-end experiments (Figure 5), we compare the performance of each parallel method with different number of tokens (from 0.5M to 4M).
>
> For your convience, we summarize the results as follows:
>
> * All methods become worse whe nthe sequence become longer, because the communication cost raises.
> * The longer the sequence is, the advantage of our method will be more obvious.

---

### Comment · Area_Chair_RnmU · 2024-11-21
**No author response yet**

Dear Submission4175 Authors,

ICLR encourages authors and reviewers to engage in asynchronous discussion up to the 26th Nov deadline. It would be good if you can post your responses to the reviews soon.

---

### Author Response · Authors · 2024-11-25
**Update pdf with fixing minors and extra experiments**

Dear ACs and reviewers,

We sincerely thank you for the time and effort in our work.

Additional experiments were updated, and we went through our work to fix some minors.

We will continue to refine our work.

Thanks,

Authors of submission 4175

---

### Meta-Review · Area_Chair_RnmU · 2024-12-17

**Metareview:**

The paper extends the idea of embedded sequence parallelism to multiple sequence dimensions, as seen in video generation models. In particular, the proposed method is motivated by the challenges presented by long sequences.

Reviewers agreed that the paper tackles an emerging problem of interest, and that the paper was clear in its technical presentation. However, discussions with Reviewer EErB revealed that the performance gains were - on video generation models - limited to shorter spatial sequence lengths, and that these shorter sequence lengths were the result of reducing the resolution of the input data; no evidence was provided to show that the accuracy loss was small enough to be acceptable. Moreover, the only multi-dimensional transformer models studied were the aforementioned video generation models. The paper's claims on generalizability to all multi-dimensional transformers (e.g. AlphaFold2, which was discussed in the rebuttal but no results were provided) and long sequence lengths (which was claimed in the abstract) are not fully substantiated by the empirical evidence.

**Additional Comments On Reviewer Discussion:**

Reviewers raised some questions about novelty, which were addressed.

Reviewer EErB raised questions about whether the communication analysis of baselines was correct, and the disagreement with authors was not fully resolved.

The most important discussion point was about the additional long spatial length results presented by the authors in their discussion with Reviewer EErB. These results showed that the communication overheads of DS-Ulysses and Megatron-LM were in the range of 1.5-2.5%, and that the proposed method reduced the communication overhead to <1%. Although the proposed method offers a reduction in overheads, these overheads were small to begin with in the baselines, which implies a negligible speedup. At best, the paper only demonstrates significant speedup over baselines on shorter sequence lengths, but this contradicts the paper's claim to be improving the state of the art for long sequences in multi-dimensional transformers.

---

### Decision · Program_Chairs · 2025-01-22

Reject